# Accumulation of Metals in the Environment and Grazing Livestock near A Mongolian Mining Area

**DOI:** 10.3390/toxics10120773

**Published:** 2022-12-10

**Authors:** Bayartogtokh Bataa, Kodai Motohira, Delgermurun Dugar, Tsend-Ayush Sainnokhoi, Lkhamjav Gendenpil, Tserenchimed Sainnokhoi, Bolormaa Pelden, Yared Beyene Yohannes, Sumiya Ganzorig, Shouta M. M. Nakayama, Mayumi Ishizuka, Yoshinori Ikenaka

**Affiliations:** 1Department of Pharmacology and Internal Medicine, School of Veterinary Medicine, Mongolian University of Life Sciences, Zaisan, Khan-Uul District, Ulaanbaatar 17024, Mongolia; 2Laboratory of Toxicology, Department of Environmental Veterinary Sciences, Hokkaido University, Kita 18, Nishi 9, Kita-ku, Sapporo 060-0818, Hokkaido, Japan; 3State Central Veterinary Laboratory, Zaisan, Khan-Uul District, Ulaanbaatar 17024, Mongolia; 4Department of Chemistry, College of Natural and Computational Science, University of Gondar, Gondar P.O. Box 196, Ethiopia; 5Spatial Analysis Laboratory, Department of Biology, National University of Mongolia, Baga Toiruu 2, Sukhbaatar District, Ulaanbaatar 14200, Mongolia; 6Biomedical Sciences Department, School of Veterinary Medicine, The University of Zambia, P.O. Box 32379, Lusaka 10101, Zambia; 7Water Research Group, Unit for Environmental Sciences and Management, Potchefstroom Campus, North-West University, Private Bag X6001, Potchefstroom 2520, South Africa; 8Translational Research Unit, Veterinary Teaching Hospital, Faculty of Veterinary Medicine, Hokkaido University, Sapporo 060-0818, Hokkaido, Japan; 9One Health Research Center, Hokkaido University, Sapporo 060-0818, Hokkaido, Japan

**Keywords:** metals, animal health, monitoring, food safety

## Abstract

The Mongolian economy is supported by rich deposits of natural resources, such as copper, coal, and gold. However, the risk of heavy metal pollution to livestock and human have been recently discussed. This research collected various samples from soil and animal (sheep, goat, horse, cow, and camel), blood and organs (kidney and liver) in the Mongolian countryside. These samples were processed, and the concentration of metals was quantified using inductively coupled plasma-mass spectrometry (ICP/MS). As previously reported, arsenic was found at high levels of accumulation in soil. Selenium is another concern, as median concentration in one area exceeded the maximum allowable level. Cadmium and selenium were found to be highly accumulated in animal kidney. This research revealed the current pollution level in Mongolia based on evaluation of soil and animals. The concentration in animals could not indicate that animals had severe effects because of heavy metal exposure. However, kidney is eaten in Mongolia, and so there is a direct connection to human health, and this research suggested the possible risks posed by each edible animal. In particular, evaluation of metals in livestock is rare in Mongolia. This result can contribute to animal and human health in Mongolian communities.

## 1. Introduction

Mongolia is a landlocked country located between Russian and China, known for its rich diversity of natural resources, including gold, silver, copper, coal, fluoride, iron, petroleum oil, and uranium [1]. Traditionally, the main economic activity in Mongolia is based on agriculture, livestock, and the mineral industry. Mongolia’s economy has been expanding through exploitation of vast mineral resources during the past 20 years. The mining sector has supported economic growth and development in Mongolia, as mining recently contributed approximately 20% of gross domestic product (GDP) and about 80% of total exports in 2017 [2,3].

However, metal contamination derived from mining and industrial activity has been observed by many groups and has been reported worldwide [4,5,6,7]. For example, the area near mining site in Zambia recorded high concentration of metals including lead and cadmium, and geographic analysis revealed that the mining activity was possible source of pollution [4]. Heavy metals represent one of the main environmental pollutants that cause adverse effects to human and animal health. These effects, such as neurotoxicity, hepatoxicity, carcinogenicity, teratogenicity, and nephrotoxicity, have also been reported [8,9]. The International Agency for Research on Cancer (IARC) has classified heavy metals such as arsenic, cadmium and nickel as Group1 (carcinogenic), and lead and cobalt as Group2B (possibly carcinogenic). While metals affect human health, many reports have also indicated heavy metal toxicity in animals. For instance, cadmium is highly toxic, affecting nearly all animal systems. Arsenic shows various toxicities with gastrointestinal and nervous symptoms in cattle and induced weight loss, reduction of milk yield, anemia, and liver and kidney damage by chronic exposure [10]. Another consideration is that heavy metals cannot be degraded through physical processes. Therefore, they persist in the animal body and environment for a long period [11].

Environmental contamination and exposure of humans to metals are practical concerns in Mongolia. Contamination with metals usually results from many different sources, such as industrial waste, chemicals used in agriculture, construction, vehicular exhaust, coal and fossil fuel combustion, and mining [8,12,13]. In Ulaanbaatar, Mongolia, survey sampling in 2007 showed high arsenic contamination in soil. The possible source of this arsenic could be coal combustion and vehicles in the city [12]. Another study of samples collected in 2019 also showed high arsenic contamination in soil in Ulaanbaatar, and arsenic and chromium have been evaluated as metals with high ecological risk [14].

In recent years, environmental problems have gained attention in Mongolia. Economic losses because of metals through toxic effects on livestock and humans have also been found in other countries [15,16]. Concerning human health, various pathways through food, air, and water were considered as intake routes for metals, and animal products are one possible source of metals to humans [8,17]. Traditionally, Mongolians eat a lot of meats, and sheep meat is one of the main protein sources. The major livestock are sheep and cows for meat and milk in Mongolia. Mongolians eat not only muscle, but also liver, kidney, heart, etc. This unique food culture may increase the risk of metal contamination of animal products. As toxicity of metals in humans has been a concern in several countries, monitoring metals in grazing livestock in Mongolia is necessary in order to protect both animal health and human food safety. However, this kind of scientific research has not been pursued in detail in Mongolia at present. Therefore, monitoring environmental pollution that can cause toxicity to animals is desirable in Mongolia. Thus, this research targeted various livestock and soils in Mongolia, analyzing metal levels in collected samples.

## 2. Materials and Methods

### 2.1. Study Area

This study was conducted on animal farms in two areas: Dornogovi (Ulaanbadrakh: UB, Zuun-bayan: ZB and Airag: AE soum) and Tuv-Aimag (Zaamar soum). Dornogovi (44°53′N 110°09′E) is one of the 21 aimags (provinces) of Mongolia, located about 456 km southeast of the capital city, Ulaanbaatar. Dornogovi is known for its harsh weather conditions; it is located in the Gobi Desert, where frequent sand- and snowstorms occur. Tuv-aimag is another province of Mongolia, located about 45 km from the capital city. Many animal products, including milk and meat, are produced in both areas and distributed in Ulaanbaatar. Another notable product in Tuv-Aimag is gold from the mining located on the Tuul River. This area is still used for mining, and new mines are continuously being developed. Dornogovi also has mining areas, for fluorspar, uranium, etc. The soil and tissue samples were collected near mining areas. Details are shown in Figure 1.

### 2.2. Sample Collection

All samplings were done in June, 2016, in Mongolia. In total, 53 tissue samples (liver and kidney) were collected from various deceased animals (horse, goat, sheep, and cow) in Dornogovi province for metal analysis. In addition, 55 soil samples were collected from the same area. Approximately 50 g of soil from a depth of 0–5 cm was collected for each sample and kept in a plastic tube. At least three composite soil samples were collected from each sampling point. Approximately 150 blood samples were randomly collected from various animals (horse, goat, sheep, cow, and camel) in Dornogovi and Tuv-Aimag. Collected tissue and blood samples were stored in cooler boxes with dry ice during the sampling on-site, and all samples were transported and kept at 4 °C for soil and at −20 °C for blood and organs at the School of Veterinary Medicine, Mongolian University of Life Sciences, before being transported and analyzed for metal concentration at the School of Veterinary Medicine, Hokkaido University, Japan around 20 days after sample collection.

### 2.3. Sample Preparation and Metal Extraction

All laboratory materials and instruments used in metal extraction were washed in 2% nitric acid (HNO_3_) and oven-dried (50 °C). Metals were extracted from the samples (tissue, soil, and blood) by acid digestion with the method used in previous reports for tissue samples [18] and environment samples [19], with minor modifications.

Tissue samples (liver and kidney) were weighed (1 g) and dried for 48 h in an oven at 60 °C. Then, the dried tissue samples were digested using 5 mL 30% nitric acid (HNO_3_) and 1 mL 30% hydrogen peroxide (H_2_O_2_) (analytical grade, Kanto Chemical Corp. Japan) in a speed wave MWS-2 microwave oven system. Digestion of blood samples (1 mL) followed the same process as the tissue samples. The microwave system was programmed to ramp in five min to 160 °C (held for five min), followed by a ramp time of three min increased to 200 °C (held for 20 min), and then hold for 10 min at 75 °C. After digestion, samples were cooled and transferred to a 15 mL polypropylene tube followed by dilution to 10 mL with Milli Q water.

In the case of soil samples, extraction was performed using about 0.5 g of sample and 5 mL 60% HNO_3_, and 1 mL 30% H_2_O_2_. The microwave was programmed at an initial temperature of 150 °C (held for 5 min), then with a ramp time of 2 min increased to 175 °C (held for 10 min) and with a ramp time of three min increased to 190 °C (held for 30 min). After digestion, samples were filtered through ADVANTEC 5C filter paper into 15 mL polypropylene tubes and made up to 10 mL using Milli Q water. Analytical blanks were run for all samples.

Tissues, blood, and soil were assessed for concentrations of ten metals (Cr, Mn, Co, Ni, Cu, Zn, As, Se, Cd, and Pb) using an inductively coupled plasma –mass spectrometer (ICP-MS; 7700 series, Agilent Technologies, Tokyo, Japan) using ^9^Be, ^115^In, and ^205^Tl as internal standards.

### 2.4. Quality Control and Quality Assurance

All chemicals and standard stock solutions were analytical reagent grade (Wako Pure Chemicals, Osaka, Japan). For quality control, blanks were analyzed after every 10 samples, and a calibration curve for each element was constructed with an R2 value greater than 0.995. Analytical quality control was performed using certified reference materials: DOLT-4 (dogfish liver, National Research Council of Canada), DORM (fish protein, National Research Council of Canada, Ottawa, Canada), SRM-1944 (New/York/New Jersey Waterway Sediment, National Institute of Standards and Technology, New York, USA), and BCR-320 (channel sediment, Community Bureau of Reference of European Commission, Brussels, Belgium). Replicate analysis of these reference materials showed good recoveries, ranging from 90–110%.

### 2.5. Statistical Analysis

JMP 16 (SAS Institute, NC, USA) software was used for all statistical analyses, with a significance threshold of *p* < 0.05. The regional difference in soils was tested using Steel-Dwass tests. Outliers were detected assessing quantile range outliers (tail quantile 0.05, Q 10). Four metal concentrations in animal blood (from two individual outlier animals) were extracted. These two animals were also defined as the first and second biggest neighborhood distance calculated from K nearest neighbor outliers (K: 2–8, threshold was not decided). Thus, these two animals were excluded from the sample dataset. There was no outlier among soil samples, which were handled in the same way as animal samples.

## 3. Results

The metal concentration in the soil is shown in Table 1 and Appendix A. The Tuv-Aimag Zaamar region had generally higher concentrations of most metals; copper and chromium concentrations were significantly higher than in other areas (*p* < 0.05) (Appendix A). Table 1 shows the median concentration of metals in the highest-concentration area, the highest concentration in this study, and the regulations (such as the maximum allowable level in Mongolia, maximum permissible level summarized in [20] and Ecological Soil Screening Levels (Eco-SSLs) [21,22,23,24,25,26,27,28]). Eco-SSLs were provided from the U.S. Environmental Protection Agency (EPA). The Eco-SSLs suggest screening the level of contaminants in soil to indicate ecological risks. Contaminants exceeding their permissible concentration may require further evaluation for protection against ecological risk. The Eco-SSLs mention four different levels of concentration (plants, soil invertebrates, birds, and mammals); this study used its levels for mammals, as shown in Table 1. Chiroma T. M et al. (2014) summarized the maximum allowable limit using World Health Organization (WHO), Food and Agricultural Organization (FAO), and Ewers, U. (1991), Standard Guidelines in Europe. Some of the highest concentrations of target metals exceeded these maximum levels. In particular, arsenic and selenium concentrations were high, and some soil samples have exceeded these levels (Figure 2). For example, more than 50% of samples in Dornogovi AE exceeded the maximum permissible level of selenium (10 mg/kg) in Mongolia.

Blood metal concentration is described in Table 2. Strong species differences were not observed for many metals in the same area. However, the camel accumulated a slightly higher concentration of zinc than the other animals in all areas. The median concentration of zinc in Dornogovi UB was 50.4 mg/kg (dry weight) in camels’ blood, while goats and sheep had blood levels of only 29.0 and 20.6 mg/kg, respectively. In the case of copper, the concentration in small ruminants, such as goat and sheep, was a bit higher than in other animals. The order of copper median concentration in Dornogovi ZB was goat > sheep > horse > camel > cow (Table 2). Comparing the regional difference showed that the Tuv-Aimag Zaamar region had a higher concentration of arsenic than other areas, as the soil concentration had already suggested. However, in the case of zinc, selenium, and lead, the median concentration in soil in Dornogovi UB was the lowest, but comparatively high concentrations of metals were detected in blood in that area, as the median zinc and lead concentration (29.0 and 0.1 mg/kg dry weight, respectively) was the highest in goat among all areas.

To give a better understanding of the absolute concentrations of metals in livestock, Appendix A summarizes the normal range of metals in animal organs, referring to published values [29]. The concentration in liver and kidney in sheep, goat, and cattle is shown in Table 3. The sample number of organs was not enough to determine regional difference. Thus, Table 3 shows mixed results from all areas (Dornogovi UB, ZB, AE). We also extracted the results of copper, zinc, and selenium in liver and kidney, as shown in Figure 3, with toxic and deficient levels described in a previous report [30]. The relatively high concentration of copper was also seen in sheep liver (median concentrations were 235.1, 92.4, and 148.3 mg/kg dry weight in sheep, goat, and cow, respectively). Arsenic concentration in organs was not as high compared with other high toxic metals, such as cadmium and lead (Table 3).

We additionally collected only two horse kidney samples, but concentrations of cadmium (122.65 and 352.79 mg/kg) in kidney were quite high in both individuals, compared with other animals, as median concentration values in sheep, goat, and cattle were 0.28, 1.02, and 3.78 mg/kg respectively (data not shown).

To summarize our results, each metal concentration was normalized using its sum and standard deviation of all the samples. Then, the mean of each normalized concentration, as shown in Appendix A, was used to evaluate the accumulation pattern of each metal.

## 4. Discussion

### 4.1. Regulations Related to Heavy Metal and Overview of Results

Table 1 shows two previous studies conducted in Mongolia. The sampling location was different from this study. This study targeted the countryside, while previous reports focused on the city [12,14]. In the current study, notable accumulation was seen in arsenic and selenium, as their concentration in some samples exceeded both Mongolian regulations and Eco-SSLs. The Mongolian regulation level was almost the same as the maximum permissible level summarized in Chiroma T. M et al. (2014), except in the cases of chromium (manganese was not evaluated in the Mongolian regulation). Thus, we used Mongolian regulations to discuss the concentration of metals in soil, as these correspond better to local people in Mongolia. Firstly, the concentration of arsenic and selenium in soil was higher than Mongolian regulation in some samples, and these metals are highly toxic. Thus, Section 4.2 and Section 4.3 mentions arsenic and selenium concentrations in Mongolia and compared with previous studies. Section 4.4 focuses on copper and zinc, which are essential metals in animal body. The concentrations of cadmium and lead were not noticeable in the soil sample. However, these metals were known as one of the most toxic metals to animal body. Thus, we mentioned the toxicities of these metals in Section 4.5 with the current results. At the end, we discuss the possible sources of heavy metal contamination in livestock and humans in Section 4.6.

### 4.2. Arsenic Contamination in Mongolia

This study showed the same trend of high arsenic concentration in soil as the results reported in other studies [12,14]. Thus, arsenic pollution could be a problem all around Mongolia. Arsenic was one of main hazardous inorganic chemical in farm animals, and many toxicities in animals were reported. Daily administration of 20% of the acute lethal dose 50 (ALD50) of arsenic for 12 weeks to goats caused body weight loss, increased respiratory and heart rate, and animal death [31]. A study targeting farm animals in an arsenic-polluted area in India showed significantly higher liver markers, such as aspartate aminotransaminase (AST) and alanine aminotransaminase (ALT), lipid peroxides, and lower superoxide dismutase (SOD), which is one of difference mechanisms against oxidative damage, compared with a control area [32]. Oxidative stress derived from arsenic has been demonstrated using experimental rodents [33]. In this study, in spite of the high concentration of arsenic in soil, its contamination in blood and organs was less than that of other toxic metals. This result followed previous research conducted in Galicia, in which the majority of sheep samples did not accumulate arsenic, even in an arsenic-contaminated area [34]. The monitoring of only one environment (soil) and animals (blood and organs) can mask the risk to animals, as the accumulation patterns of metals and metalloids can be different, depending on the metals (Appendix A). The lack of usefulness of blood arsenic concentration as a biomarker for long-term exposure to arsenic has already been reported; this is due to its earlier excretion from the blood compared to other metals and metalloids [35]. Thus, investigating both animals and environments would be beneficial to understand the effects of metals on livestock.

### 4.3. Selenium Contamination in Mongolia

This study revealed similar concentrations of metals to those described by a previous study conducted in Mongolia examining many metals; however, selenium concentration was high in soil in this study (Table 1). Selenium is distributed variably in the world, depending on geological conditions. Selenium is an essential element for both human and animal bodies, but there is a narrow margin between toxic and deficient levels [36]. For instance, high doses of selenium in lambs were associated with myocardial necrosis, pulmonary edema, and hemorrhage in one histopathological study [37]. An exposure experiment conducted using buffalo calves found that adverse effects and mortality were observed when blood concentrations of selenium became more than 2 and 3.4 µg/mL, respectively. That study investigated relatively acute toxicity during a 30-day observation [38]. Extrapolation to the current study should be done carefully, but the maximum selenium concentration found in this study was around 0.44 µg/mL. A previous review suggested that a selenium concentration between 1.25 and 2.5 mg/kg in the liver is adequate level for cattle, and between 2.3 and 8.0 mg/kg dry weight for newborn calves [30]. The median concentration in sheep and goat was around 2.0 mg/kg, which is an adequate level in liver in this study (Figure 3). Another review mentioned that selenium levels of more than 2 ppm in liver caused acute toxicity, and in many cases reported that sheep died due to acute/subacute intoxication [36]. We did not use the wet weight for calculation because we collected liver samples from the dead body, and conditions such as water content can vary depending on when and where animals have died. However, considering the normal water content in liver (around 3.5—4 times higher concentration in dry weight than that of wet weight [29]), most liver samples would have levels of selenium less than 2 ppm. Thus, even though the selenium concentration in soil exceeded Mongolian regulation levels, we cannot conclude that animals showed an intolerable toxic effect from selenium in Mongolia. However, in cows, the median concentration was higher than in small ruminants. The concentration in three out of four cows exceeded adequate levels. Monitoring selenium in cows is thus required.

### 4.4. Copper and Zinc Contamination in Mongolia

Copper and zinc are essential elements for the animal body [39]. Thus, the high absolute concentrations of these metals in animal blood, liver, and kidney in this study were reasonable. Susceptibility to copper is higher in sheep > cattle > monogastric animals [40]. The results indicate that susceptible animals were exposed to copper more frequently. The season and hemoglobin type differentiate copper concentrations in sheep. Thus, simple comparison was difficult, but the copper concentration in blood in the current study was relatively low compared with other previous studies [41]. In this area, copper deficiency might be a risk to animals, rather than intoxication. When focusing on liver concentration, previous review report summarized that the cutoff for deficient and adequate copper concentration in liver would be less than 33 and more than 125 mg/kg dry weight, respectively [30]. Most individuals collected in this study demonstrated more than a deficient level (Figure 3). The same report mentioned that copper at a level more than 1250 mg/kg is toxic in cattle liver, but no animal exceeded this value in this study. In the case of zinc, most samples ranged between adequate and high accumulation level. The toxic level (more than 1000 mg/kg in cattle) was not seen in the current dataset (Figure 3) [30] and the risk related to zinc would be relatively low in the current sampling area. Both copper and zinc had broad range of concentration from the lowest to the highest in samples. We did not have their ages data, and this is our limitation. For more proper diagnosis of deficiency, the sampling of liver and kidney from not only dead body but also from healthy slaughter animals with their enough information such as age are needed. Overall, considering the concentration of copper and zinc in blood and animal organs, the risk of intoxication was relatively low compared with arsenic and selenium.

However, zinc concentration in camel blood was high in two areas where we took camel samples. An exposure study in cows showed significant induction of the enzymes which have protective effects against oxidative stress such as SOD at around 2.5—3.0 ug/mL blood zinc concentration [42]. The current study revealed that some camels had more than 5.0 ug/mL of zinc in their blood (data not shown, Table 2 shows only dry weight). Camels are a unique animal living in a severe desert environment. The camel has a unique antioxidant defense system that uses copper; zinc superoxide dismutase was also found [43]. This mechanism allows a high concentration of zinc in camels. However, a previous report also showed that camels maintain a low plasma zinc concentration compared with cows, even when the two species ate similar feed containing multiple metals [44], contrary to the current study. Reports describing camels are fewer in number than those describing other livestock. Further study and measurement of blood parameters, including oxidative stress, in camel are needed to evaluate the effect of metals on camels in this area.

### 4.5. Cadmium and Lead Contamination in Mongolia

The median concentration of lead and cadmium in soil was less than levels allowed by Mongolian regulation. However, values exceeded Eco-SSLs in some samples. These metals are well known to have high toxicity to humans. The high concentration of cadmium in horses was derived from its high accumulative ability in these animals [45]. The previous report showed much higher concentration of cadmium in horses than in other farm animals, even from the same area [46]. The concentration in horse kidney in this study also incommensurably high compared with other animals in same area. This study has only two kidney samples in horse, but the result might provide hints for communities in Mongolia to carefully consider the use of horse kidneys as food. Itai-itai disease was a notorious incident resulting from cadmium pollution, causing severe symptoms in humans in Japan [47]. The cadmium concentration in soil was quite low in Mongolia, and most of samples had less than 0.1 mg/kg in this study. However, cadmium is highly accumulated in cow kidneys (median concentration: 3.8 mg/kg). Lead also induces severe symptoms in mammals. For instance, previous in vivo Pb exposure study on mice and environmental Pb exposure effect on human showed inhibited activity of d-aminolevulinic acid dehydratase (d-ALAD), a porphobilinogen synthase enzyme which is responsible for heme synthesis [48,49]. The activity of ALAD was estimated to be inhibited by lead with less than around 0.55 mg/kg dry weight in cattle [50]. When focusing on ALAD activity, the cattle in the current sampling area would not get a significant effect of lead as the highest median concentration in cattle was 0.15 mg/kg in blood in Tuv-Aimag Zaamar. However, in particular, cadmium and lead induce severe toxicity to mammals at low concentrations [51]. Thus, even if the detected concentration was less than allowed by Mongolian regulation, we should continue to monitor these metals in order to secure good animal health.

### 4.6. Source of Heavy Metal Contamination into Livestock and Human

When comparing regional differences in selenium accumulation, the Tuv-Aimag Zaamar area was noticeably lower than other areas, as median concentration in goat and sheep was 0.27 and 0.26 mg/kg in blood, respectively (Table 2). However, the soil accumulated selenium equally between Dornogovi and Tuv-Aimag Zaamar (Appendix A). As a result, other pathways for heavy metals to reach livestock must also be considered. One of the important sources of heavy metal in livestock is water [10]. For example, a report from India revealed that livestock drinking water containing mercury and manganese at levels higher than their permissible limits. The relationship of metals in livestock to metals in their feed has been also frequently reported, as feed is a major pathway of toxic metal intake for livestock [15,52]. By free grazing, which is the usual form in the Mongolian countryside, sheep and goats eat wild plants. The regional difference in selenium in animal blood could be derived from the local plants. Unusual contamination patterns were also found, such as an animal living in a low-contamination area (Dornogovi UB) accumulating a relatively high concentration of zinc and lead in its blood. As a result of these contaminations, the animals may suffer from effects of the accumulation of heavy metals in their feed (wild plants) and drinking water. Species differences can cause different accumulation patterns of metals in plant [53]. Thus, further investigation of vegetation at sampling sites and metal accumulation in livestock drinking water and plants would be needed to identify the source of contaminants in animals.

Both soil and animal products (liver and kidney) can be sources of metal contamination in human. Previous research monitored the heavy metal concentration in soil in Mongolia and alerted the community to the possible risk on human health [14]. In addition, this research will contribute to evaluation of the impact of metals on human health, particularly involving animal products. The results in this study clearly showed the difference of accumulation pattern of metals among animal and soil samples. For example, arsenic and chromium were much higher in soil than animal samples (Appendix A). Even between liver and kidney, for example, selenium and cadmium preferably accumulated in kidney. This might help the identification of possible sources of metals in human. However, for the evaluation of possible risk in human, the daily intake of kidney and liver in Mongolia is necessary, but unfortunately these kinds of information were not found. Thus, further investigation of liver and kidney intake is desirable.

## 5. Conclusions

The international scientific report targeting metals in Mongolia was limited to the veterinary field. This paper revealed that arsenic and selenium were highly accumulated in Mongolian soil. They should be considered carefully as a risk, but the current concentration results in animal organs were not enough to conclude that animals suffer severe adverse effects because of exposure to these metals. This research also indicates that screening of all of soils, blood, and organs can be effective to monitor metals and to understand the inclusive polluted levels in Mongolia, as arsenic highly accumulated in soil but not in animals. For a better understanding of sources of heavy metal contamination, the monitoring of heavy metals in wild plant and livestock drinking water is desirable. We also focused on animal organs (liver and kidney), which are eaten in the countryside in Mongolia. Metals can be transferred to humans via animal products. Our results contribute to the protection of not only animal health, but also human food safety. This research will be beneficial to protect animal health, inform future studies targeting humans, and help reveal the source of metal contamination.

## Figures and Tables

**Figure 1 toxics-10-00773-f001:**
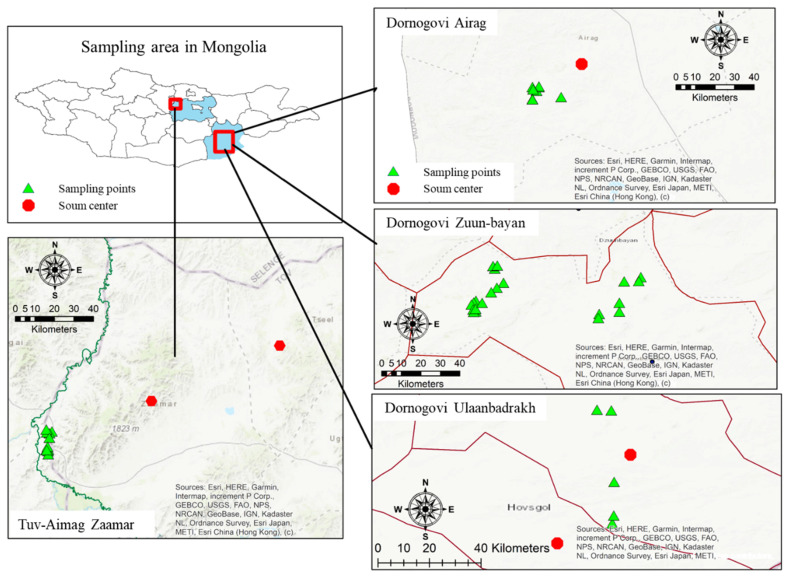
Sampling location. Sampling areas in Tuv-Aimag Zaamar, Dornogovi Airag, Zuun-bayan and Ulaanbadrakh were described by map using ArcGIS 10.7.1 (ESRI Co., Redlands, CA, USA).

**Figure 2 toxics-10-00773-f002:**
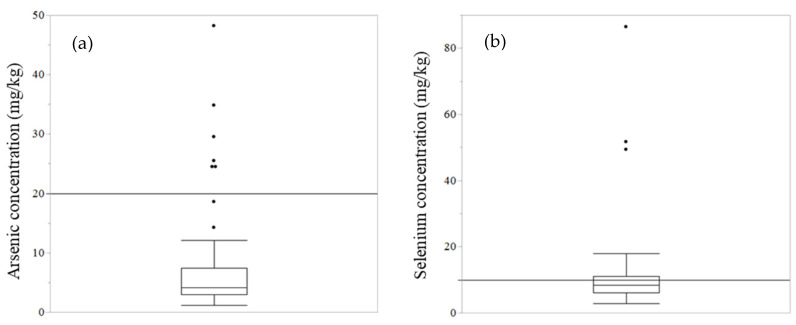
Arsenic and selenium concentration in soil (all samples in this study). Arsenic (**a**) and selenium (**b**) concentration in all soil samples are shown in a box plot. The black line shows the maximum allowable limit, as regulated in Mongolia (arsenic: 20 mg/kg soil, selenium: 10 mg/kg soil).

**Figure 3 toxics-10-00773-f003:**
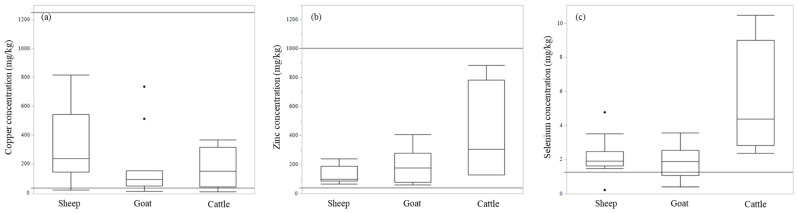
Copper, zinc and selenium concentration in liver. Copper (**a**), zinc (**b**) and selenium (**c**) concentration in liver samples are shown in a box plot. The black line shows toxic and deficient level in (**a**) and (**b**), and marginally deficient level in (**c**). Deficient lines in (**b**) and (**c**) indicated 40 and 1.25 mg/kg although they had range 20—40 and 0.6—1.25 mg/kg respectively.

**Table 1 toxics-10-00773-t001:** Metal concentration in soil in this study, previous study in Mongolia, and maximum allowable limits.

	Mongolian Government Limit (mg/kg)	Eco-SSLs (mg/kg)	Maximum Permissible Level (mg/kg) *	This Study (2016) with Recorded Area (mg/kg)			Previous Study in Mongolia (2019) (mg/kg)	Previous Study in Mongolia (2007) (mg/kg)
Cr	150	Ⅲ 34	100	Median	26.5	(Tuv aimag)	29	20.3
		Ⅳ 130		Highest	120.6	(Dornogovi ZB)	110.4	23.9
Mn		4000	2000	Median	377.8	(Tuv aimag)	0.5	
				Highest	3504.0	(Dornogovi ZB)	1.1	
Cu	100	49	100	Median	18.1	(Tuv aimag)	28.9	35.9
				Highest	64.7	(Dornogovi ZB)	78.6	47.4
Zn	300	79	300	Median	34.5	(Tuv aimag)	135.6	158.7
				Highest	209.1	(Dornogovi ZB)	384	203.0
As	20	46	20	Median	8.8	(Tuv aimag)	22.9	14.0
				Highest	48.2	(Dornogovi ZB)	526.8	64.1
Se	10	0.63	10	Median	10.7	(Dornogovi AE)	0.1	
				Highest	86.5	(Dornogovi ZB)	1.5	
Cd	3	0.36	3	Median	0.07	(Tuv aimag)	0.2	0.8
				Highest	0.44	(Dornogovi ZB)	1.9	1.1
Pb	100	56	100	Median	7.5	(Tuv aimag)	34.5	63.9
				Highest	76.5	(Dornogovi ZB)	77.3	119.6

The metal concentration in this study is shown with results of a previous study conducted in Mongolia. Brackets indicate the sampling year in each study. * is the reported maximum allowable level summarized in Chiroma T. M et al. (2014). All concentrations are described in mg/kg.

**Table 2 toxics-10-00773-t002:** Metal concentration in blood.

Dornogovi AE		Cr			Mn			Cu			Zn			As			Se			Cd			Pb		
Goat (15)	median	0.05			0.074			4.0			11.4			0.011			0.95			< 0.001			0.077		
(mg/kg)	range	0.023	—	0.14	0.046	—	0.31	1.6	—	4.8	6.8	—	18.3	0.009	—	0.023	0.47	—	1.4	< 0.001	—	< 0.001	0.036	—	0.13
Sheep (10)	median	0.0525			0.11			3.5			10.0			0.0125			1.2			< 0.001			0.055		
(mg/kg)	range	0.031	—	0.075	0.078	—	0.15	2.4	—	4.5	6.4	—	14.6	0.009	—	0.015	0.59	—	1.5	< 0.001	—	< 0.001	0.03	—	0.088
Dornogovi UB		Cr			Mn			Cu			Zn			As			Se			Cd			Pb		
Goat (10)	median	0.045			0.10			5.2			29.0			0.013			1.5			0.001			0.10		
(mg/kg)	range	0.031	—	0.20	0.069	—	0.37	4.7	—	8.1	9.4	—	148.1	0.008	—	0.049	1.2	—	1.9	< 0.001	—	0.003	0.033	—	0.22
Sheep (10)	median	0.054			0.13			4.9			20.6			0.016			1.6			0.001			0.095		
(mg/kg)	range	0.032	—	0.15	0.085	—	0.24	3.7	—	7.9	8.9	—	63.4	0.011	—	0.020	1.3	—	2.2	< 0.001	—	0.004	0.038	—	0.14
Horse (3)	median	0.055			0.09			3.7			9.3			0.013			0.85			0.003			0.036		
(mg/kg)	range	0.052	—	0.12	0.067	—	0.12	2.8	—	6.7	7.0	—	10.1	0.008	—	0.013	0.84	—	1.0	0.003	—	0.003	0.031	—	0.039
Camel (13)	median	0.035			0.20			2.9			50.4			0.01			1.4			< 0.001			0.058		
(mg/kg)	range	0.015	—	0.18	0.078	—	0.80	1.8	—	19.2	15.1	—	211.2	0.005	—	0.072	0.88	—	10.8	< 0.001	—	0.004	0.038	—	0.37
Dornogovi ZB		Cr			Mn			Cu			Zn			As			Se			Cd			Pb		
Goat (21)	median	0.041			0.07			4.8			13.3			0.012			0.96			< 0.001			0.062		
(mg/kg)	range	0.014	—	0.14	0.017	—	0.35	2.0	—	8.4	5.2	—	62.7	0.003	—	0.063	0.54	—	2.7	< 0.001	—	0.004	0.018	—	0.19
Sheep (20)	median	0.040			0.12			4.5			11.3			0.019			2.3			0.001			0.055		
(mg/kg)	range	0.019	—	0.10	0.073	—	0.29	2.9	—	6.0	7.4	—	56.4	0.008	—	0.050	1.2	—	3.4	< 0.001	—	0.003	0.026	—	0.15
Horse (10)	median	0.044			0.11			3.6			9.4			0.015			2.0			0.002			0.047		
(mg/kg)	range	0.015	—	0.12	0.027	—	0.31	1.3	—	6.5	3.7	—	14.0	0.003	—	0.037	0.60	—	3.4	< 0.001	—	0.005	0.019	—	0.33
Camel (15)	median	0.037			0.17			2.8			31.1			0.012			1.2			0.001			0.044		
(mg/kg)	range	0.009	—	0.080	0.015	—	0.49	1.0	—	4.2	5.9	—	59.5	0.002	—	0.029	0.47	—	4.0	< 0.001	—	0.003	0.017	—	0.16
Cow (3)	median	0.026			0.04			1.4			2.6			0.003			0.62			< 0.001			0.024		
(mg/kg)	range	0.015	—	0.031	0.018	—	0.058	0.028	—	2.5	0.21	—	7.7	0.001	—	0.006	0.004	—	1.1	< 0.001	—	0.002	0.019	—	0.025
Tuv Zaamar		Cr			Mn			Cu			Zn			As			Se			Cd			Pb		
Goat (9)	median	0.053			0.13			5.5			21.1			0.047			0.27			0.001			0.065		
(mg/kg)	range	0.032	—	0.089	0.059	—	0.32	4.4	—	6.3	8.0	—	44.9	0.007	—	0.094	0.18	—	0.41	< 0.001	—	0.005	0.042	—	0.13
Sheep (10)	median	0.044			0.13			4.4			15.7			0.042			0.26			< 0.001			0.052		
(mg/kg)	range	0.021	—	0.17	0.056	—	0.28	1.9	—	5.0	4.3	—	32.4	0.004	—	0.079	0.076	—	0.42	< 0.001	—	0.004	0.015	—	0.18
Cow (5)	median	0.068			0.10			4.2			34.0			0.078			0.14			0.002			0.15		
(mg/kg)	range	0.042	—	0.079	0.063	—	0.25	4.0	—	4.7	23.6	—	101.2	0.067	—	0.39	0.11	—	0.18	0.001	—	0.005	0.091	—	0.21

Metal concentration is shown, separated by region and animal species. All concentrations are given in mg/kg (dry weight). The number of samples in each area is indicated in parentheses.

**Table 3 toxics-10-00773-t003:** Metal concentration in liver (3a) and kidney (3b).

(3a)		Cr			Mn			Cu			Zn			As			Se			Cd			Pb		
Sheep (10) (mg/kg)	median	0.21			12.4			235.1			99.5			0.06			1.9			0.28			0.38		
	range	0.03	—	0.57	3.7	—	26.6	20.8	—	814.9	65.0	—	239.1	0.03	—	0.16	0.22	—	4.8	0.002	—	0.83	0.22	—	2.2
Goat (11) (mg/kg)	median	0.28			10.2			92.4			176.2			0.06			1.9			0.11			0.24		
	range	0.07	—	1.1	4.0	—	33.4	10.5	—	734.5	59.9	—	406.6	0.01	—	0.27	0.40	—	3.6	0.002	—	0.72	0.08	—	2.8
Cattle (4) (mg/kg)	median	0.28			16.2			148.3			305.0			0.19			4.4			1.02			0.31		
	range	0.11	—	0.60	2.7	—	32.9	8.8	—	365.5	127.8	—	885.9	0.06	—	0.42	2.4	—	10.5	0.320	—	1.8	0.20	—	0.35
(3b)		Cr			Mn			Cu			Zn			As			Se			Cd			Pb		
Sheep (8) (mg/kg)	median	0.27			3.9			16.6			88.8			0.07			4.3			0.28			0.21		
	range	0.10	—	0.52	2.5	—	5.9	8.7	—	43.8	45.9	—	231.2	0.03	—	0.13	1.8	—	11.9	0.003	—	4.1	0.10	—	0.67
Goat (11) (mg/kg)	median	0.29			4.8			15.0			87.7			0.07			5.0			1.02			0.24		
	range	0.08	—	0.61	1.9	—	22.3	8.4	—	279.1	49.3	—	777.1	0.01	—	0.27	1.1	—	9.6	0.004	—	7.0	0.06	—	0.99
Cattle (4) (mg/kg)	median	0.32			4.4			39.2			306.4			0.24			8.2			3.8			0.60		
	range	0.18	—	2.2	4.3	—	6.3	11.6	—	99.8	95.3	—	408.5	0.15	—	0.38	7.4	—	10.1	0.99	—	6.9	0.25	—	1.1

Metal concentration in liver and kidney is shown. All concentrations are given in mg/kg (dry weight). The number of samples in each area is indicated in parentheses.

## Data Availability

The datasets used in the study are available in the Appendix A at https://www.mdpi.com/article/10.3390/toxics10120773/s1 and from the corresponding author on reasonable request.

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
