# Peer review of "Accumulation of Metals in the Environment and Grazing Livestock near A Mongolian Mining Area"

_toxics, 2022, doi:10.3390/toxics10120773_

Round 1

Reviewer 1 Report

The submitted manuscript for review deals with the heavy metal content of soils, blood and various animal organs in rural Mongolia, and on this basis an attempt was made to assess environmental risks. The study revealed the current level of pollution in Mongolia based on soil and animal assessments. The authors suggested possible risks posed by each edible animal, taking into account species differences. In particular, it is significant that an assessment of the metal content of livestock was conducted, which is very rarely done in Mongolia. By assessing contamination, these results can contribute to the health of animals and humans in Mongolian communities and extending them to other Asian countries can contribute to the overall assessment of pollution and exposure of the area's population.

The study collected various samples from soil and blood from animals (sheep, goats, horses, cows and camels) and organs (kidneys and livers) in the Mongolian countryside. These samples were processed, and metal concentrations were quantified using inductively coupled plasma mass spectrometry (ICP/MS). Unfortunately, the methodology needs to be completed and explained. E.g. Why were different conditions and a more concentrated nitric acid used for the blood sample dilution than for tissue and soil?

In the work, the spelling of the sum formulas of chemical compounds should be corrected, the lack of units in the tables should be completed.

Tables in the ESI should contain the letter S or something like it, so that it is clearly separated between the main manuscript and supplementary materials. In addition, the tables should have a uniform format so as not to hinder the interpretation of the results.

Why are the results in Table 1 and Table 1 in the supplementary material completely different?

What do the numbers in parentheses (9; 8; 15; 23) in Table 1 in the supplementary material mean?

The methodology should also be supplemented with how the soil was stored for analysis, at what time from sampling the analysis was performed, at what temperature the soil was transported and stored before analysis.

Author Response

Thank you very much for your kind comments and suggestions.

Reviewer 2 Report

The article contains interesting data but presentation and discussion of this data have to be improved. 

First of all, the bulky Tables spoil the impression. You'd better to present this information as Figures.

Second, you missed discussion of some metals behaviour. It make a reader to analyze large amount of data from the bulky Tables.

Third, you should review the results of PCA, explain and discuss them in proper way.

Specific comments see in the pdf file.

Author Response

(The authors gave the same response as above.)

Round 2

Reviewer 1 Report

I have no comments on the submitted completed version of the manuscript. I think the work looks much better after the changed.

Author Response

Thank you very much for evaluating our manuscript.

Reviewer 2 Report

Supplementary materials now have all necessary Figs but I still recommend to delete PCA analysis from the article, because no appropriate interpretation was done in the article

I strongly recommend to delete PCA analysis from the article, because no appropriate interpretation was done.

Author Response

Thank you very much for your suggestion. We deleted Supplementary Figure S1 and deleted or regulated related explanations in the main text.